# Innovative Controllable Torsional Damper Based on Vacuum Packed Particles

**DOI:** 10.3390/ma13194356

**Published:** 2020-09-30

**Authors:** Dominik Rodak, Robert Zalewski

**Affiliations:** Institute of Machines Design Fundamentals, Warsaw University of Technology, 02-254 Warsaw, Poland; Robert.Zalewski@pw.edu.pl

**Keywords:** vacuum packed particles, torsional vibration damping, smart materials

## Abstract

In this paper a new concept of a controllable granular damper is presented. The introduced prototype works based on so-called vacuum packed particles (VPPs). Such structures are made of granular materials located in a soft and hermetic encapsulation. As a result of generating a partial vacuum inside the system, the structure starts to behave like a nonclassical solid body. The global physical (mechanical) features of VPPs depend on the level of internal underpressure. The introduced prototype of a controllable torsional damper exhibits various dissipative properties as a function of internal underpressure. The design details of the investigated device are presented. Basic laboratory tests results are discussed. To describe the hysteretic behavior of the device, the Bouc–Wen rheological model has been modified and adopted. Nonlinear functions of underpressure have been introduced to the initial model formulation. The developed Bouc–Wen model has been applied to capture the real response of the VPP torsional damper prototype.

## 1. Introduction

Damping of torsional vibrations is a highly important and challenging problem in many fields of industry. Protection from the negative impact of vibrations is one of the basic ways to extend machine life. Rotors, shafts and other rotational parts operate under a certain range of excitation frequencies, close to the natural frequency of a structure. These may cause a dangerous intensification of amplitudes, because of the resonance phenomenon. Damping of random excitation, noise reduction and preventing people from hazardous vibrations should be continuous motivations for development of innovative damping methods [1,2]. It is confirmed in [3] that application of intelligent dampers is recommended everywhere where such devices could be properly controlled. A parameter variation of the device is a crucial point in the control process of active and adaptive tuned mass dampers [4]. Various methods of control techniques are possible. The first group of methods involves a controlled change of the stiffness parameter of the device. Another one allows the user to change the dynamic response of the system by changing its inertial parameters. Such methods based on controlling the dynamic features of the structure can be commonly encountered in the world literature [5,6]. A different way to attenuate vibrations of the system is to dissipate its energy by changing its damping properties in real-time [7].

Popular intelligent materials which are often applied to adaptive torsional dampers are magnetorheological fluids (MRF) and magnetorheological elastomers (MRE) [8]. The shape of MREs is more specified than that of MRFs [9]. Moreover, MREs do not require hermetic spaces or limited containers [10]. To change their physical parameters such as shear modulus an external magnetic field is required [11]. Similar to other smart materials, MREs are applied to various types of structures. It is possible to find MRFs and MREs as a base of linear dampers, torsional dampers, or as a core in sandwich beams [12,13,14].

Recently, a new group of controllable structures called vacuum packed particles (VPPs) has attracted the interest of a wider society of researchers. Up until now, these structures were considered as linear dampers or multi-axial shock absorbers [15].

In this paper the authors propose an original and novel approach in the application of VPPs: the controllable torsional damper. The important feature of the VPP torsional damper prototype is a hysteretic behavior. The investigated device provides new opportunities in a semi-active damping of torsional vibrations strategies. The paper presents preliminary experimental results which lead to determination of basic mechanical properties of the discussed damper in the case of rotary movement.

In Section 1, basic information about active vibration damping is provided. In Section 2, the vacuum packed particles and their features are described. In the next section, the operating principles of the proposed device are presented and design details are depicted. Moreover, the research plan and details of experiments are provided. Quasi-static experiments for various underpressure values were carried out to collect original empirical data. Experimental results and estimated physical parameters are shown at the end of Section 3. In Section 4, the modified pressure dependent Bouc–Wen hysteresis model is introduced. In Section 5, obtained results are discussed. Experimental data are the basis for the model identification process. In the final stage of research, the real response of the prototype is verified with the numerical simulation results. Parameter characteristics are depicted in the Section 5. Conclusions are provided in the final section of the paper.

## 2. Vacuum Packed Particles

Vacuum packed particles (VPPs) are innovative structures pretending to be a group of materials, known as smart structures. VPPs are based on a loose material enclosed in a soft and hermetic envelope. When the pressure inside is close to atmospheric pressure, grains behave similarly to a dense and viscous fluid. When the partial vacuum is generated inside the structure, particles begin to come into contact with each other and with the flexible membrane separating them from the surroundings. This phenomenon strongly influences the macroscopic features of VPPs. In fact, the system is transformed into the semisolid state [16,17]. The value of underpressure (defined as the difference between atmospheric and internal pressures) and a type of granular material are parameters strongly affecting the stiffness and damping properties of the structure [18]. Vacuum packed particles exhibit several similarities to MRFs or MREs and can be modeled with similar rheological models [19,20].

An additional advantage of structures based on VPPs is that they can be customized to various external shapes. When the zero value underpressure (atmospheric pressure) is generated inside the system, VPPs can be treated as a kind of nonclassical plasticine, which easily allows the user to form any external shapes with it [17,21].

Thanks to the so-called jamming mechanism caused by the air being pumped out from the structure, the temporary shape of the VPPs can be permanently frozen [17]. Nowadays VPPs encounter many interesting practical applications.

For example, in [22] they were used as vacuum mattresses enabling injured patients to be safely transported to hospital. Another medical application of considered special granular structures can be found in controllable flexible endoscopes [23]. In the considered application, VPPs were used to design a special guide with variable rigidity, which facilitates endoscopic examinations that are not troublesome for the patient.

Vacuum packed particles are also applied in numerous controllable damping solutions. As an alternative to MRFs, VPPs are especially interesting as a basis of semi-active damping devices. Applying VPPs as a core in sandwich beams is an encouraging way to control their dynamics. In the paper [24], four different granular materials and classical rheological models were examined. Investigating the sandwich beam free vibrations, the authors showed the efficiency and accuracy of discussed models and characterized that the damping behavior depended on the grain type. In [25] or [26], free vibrations of a beam with a controllable core were examined. Presented results revealed the influence of underpressure on damping performance and stiffness. Experimental and numerical data confirmed that the logarithmic damping decrement depends on the underpressure value and increases with it. Both papers claim that VPPs are an interesting alternative for other, more sophisticated and expensive smart materials. In [25], the authors proved that the proper control of the granular core provides a significant reduction of amplitudes in the case of excited vibrations. The paper shows that the dynamic structure of a sandwich beam with a controllable granular core changes due to various values of underpressure.

It is also possible to find an application of VPPs as a linear damper of symmetrical and nonsymmetrical characteristics. According to the solution discussed in [21], the linear VPP vibration attenuator has a cylindrical shape. The single granular core is placed inside a steel spring that provides stiffness to the system. The controllable granular damper prototype was subjected to the sine excitation rule with various frequencies and values of partial vacuum. It was shown that the underpressure parameter is a suitable factor enabling the user to change the dissipative properties of the device. The biggest disadvantage of the presented VPP damper seems to be its nonsymmetrical damping characteristic. This fact means that the applicability of the discussed solution may be limited. To capture the recorded nonsymmetrical hysteresis loops, authors have proposed the modified Bouc–Wen model [27]. A similar topic of the Bouc–Wen model adaptation for cylindrical granular cores was presented in [28]. Using numerical methods for parameter identification the model was calibrated and verified with the real response. It can be concluded that typical mathematical models developed for magnetorheological (MR) fluids or MR elastomers are capable of capturing the nonlinear behavior of VPPs [28].

## 3. Experiments

### 3.1. VPP Torsional Damper Prototype

Experiments were conducted on the specially prepared VPP torsional damper prototype designed mainly using fused deposition modeling (FDM) technology. The scheme of the damper is depicted in Figure 1. It consists of the outer driver (I), the inner driver (II), two outer rings (III), two inner rings (IV), an outer flange (V), two sealing membranes (VI), two O-rings (VII) and a working chamber with granular material (VIII). The main part of the damper is an air-tight operating chamber built by the outer driver (I), inner driver (II) and sealing membranes (VI). It is filled with loose granular material (VIII). The air is pumped out of the chamber through the holes in the inner driver (II), causing suction of the membranes and change in the granular core macrostructure. The value of underpressure is controlled by a vacuum pump. The air is removed from the system through a hollow drive shaft mounted in the inner driver (II). It is possible to use various methods to drive the damper. In the discussed prototype, the rotary motion from the drive shaft is transmitted through a clamping sleeve fixed in the inner driver (II), through the deformable working chamber (VIII), to the outer driver (I), which is connected to the flange (V). The flange (V) enables the connection of the damper to the output shaft.

The detailed CAD model of the discussed torsional VPP damper is depicted in Figure 2.

The granular material applied in experiments was a semi-finished product for an injection molding technology. It was made of ABS (acrylonitrile butadiene styrene) grains. Such a material was inexpensive and commonly available. The shape of a single grain was cylindrical and dimensions were 3 mm for length and 2.5 mm for diameter. The working chamber volume was 540 cm^3^ with a filling ratio of 74.4% (426 g of grains). The filling ratio was taken empirically and was a constant parameter in all conducted tests.

### 3.2. Research Plan

The main objective of the laboratory tests was to determine the basic dissipative properties of the innovative torsional vibration attenuator as a function of the internal underpressure. This paper presents the results of the quasi-static torsion tests. All experiments were carried out with the same range of applied torsion angles (Φ = ± 5°) (Figure 3) to ensure nondestructive deformations of the external wrap and to analyze mainly the underpressure influence on dissipative characteristics of the damper. Exceeding the threshold value of the angular loading would lead to elastic (in the first stage) and plastic deformations of the external encapsulation. The plastic strains of the envelope are underpressure independent and are unwanted phenomena in VPP devices. The range of underpressure values was the same for all tests, from 0.00 to 0.05 MPa with a step of 0.005 MPa. A higher underpressure intensifies the jamming mechanism, which is connected to strong adherence of grains to the envelope. In such a case the loading caused micro perforations of the external wrap and it was difficult to control the underpressure parameter. Three loading cycles were applied to achieve a stabilized response of the device. It is worth mentioning that the loading cycles applied in the special granular structure experimental research excitation program (Figure 3) did not introduce inertial forces into the overall system, except at points where the velocity changed direction. This allowed for an accurate measurement of the torque response. The loading velocity was set as 0.5 deg/s. The strain rate was 0.0014 1/s. The recorded results were in the form of torque-angle of rotation characteristics (M = f(Φ)). Based on experimental data, it is possible to present hysteresis loops for various underpressure values. Such characteristics reflect controllable dissipative properties of the investigated VPP device.

### 3.3. Test Stand

Laboratory tests were carried out on the specially designed experimental stand. The test stand and its details are depicted in Figure 4. To provide an appropriate stiffness, the test stand consisted of aluminum alloy profiles. The torsional damper prototype was fixed in the bottom of the mounting set and connected with the encoder and the torque transducer in a way that allowed for rotation. The load was applied manually to the VPP damper by a lever (6) which rotated in the axis of the damper. The applied angles of rotation were limited by two pin-type limiters mounted on the plate (8) to keep the range of excitations constant. Signals from the torque transducer (4) and the encoder (2) were collected by a measuring card. Parts (1) and (5) were connection shafts which were coupled with sensors thanks to coupling sleeves (3). The whole measuring set was protected by the thrust washer (7) and fastened by mounting plates (8) and (9). The signal from the torque transducer was additionally amplified. All data was recorded on the PC thanks to the data acquisition system. The partial vacuum was generated by the external vacuum pump. The underpressure value was controlled by the potentiometer built into the pump. The torque was a resultant value of the assumed rotation range.

Figure 5 depicts the complete test stand. It consisted of the PC (1), the vacuum pump with the controlling valve (2), the data acquisition card (3), the signal amplifier (4) and the investigated VPP damper (5). The load lever (6), coupling sleeves (7), torque transducer (8) and encoder (9) are also shown in Figure 5.

### 3.4. Results

A single measurement cycle included mounting the damper prototype in the test bench holder, setting the appropriate partial vacuum value, loading the damper and recording the device response. Three separate measurement series were conducted to eliminate possible fatal errors. All data were collected without any additional filters.

Typical experimental dissipative characteristics (torque vs. torsion angle) for representative values of the partial vacuum are depicted in Figure 6 and Figure 7. The influence of the underpressure value on the character of recorded characteristics is evident.

The first observation based on analysis of the data presented in Figure 6 and Figure 7 is the dependence of the dissipative characteristics on the partial vacuum value generated inside the damper prototype. Torque–torsion angle characteristics are located higher for experiments conducted with greater underpressure values (Figure 6). Moreover, increased values of investigated parameters (equivalent stiffness and amount of dissipated energy) for higher partial vacuums have been observed. For the 0.00 MPa probe (atmospheric pressure), recorded values are noisy. In the case of no underpressure, membranes do not have stable contact with grains and they can easily reorganize. Small values of underpressure, like 0.01 MPa, provide more predictable behavior. Recorded hysteresis loops have nonclassical shapes because of the possible reorganization of grains and small clearances in the mounting point. This could be a reason for extension of the shape of the dissipative characteristics around the point 0.0.

Loading-slope inclination increases following an exponential trend. The difference between 0.01 and 0.02 MPa probes is more evident than between 0.02 and 0.05 MPa. The inclination of the quasi-linear part of the 0.02 MPa test result is nearly the same as for the 0.05 MPa probe. Shapes of 0.02 and 0.05 MPa damping characteristics are similar to each other. The most substantial parameter revealing the impact of a partial vacuum is the recorded maximum torque value. This parameter reaches the highest value for the 0.05 MPa probe. The amount of dissipated energy is comparable for 0.02 and 0.05 MPa probes. This fact may lead to the conclusion that from 0.02 MPa, dissipative and stiffness properties do not change significantly.

In this paper the authors particularly focused on the impact of underpressure on three parameters: the VPP damper equivalent torsional stiffness (k_eq_a_), maximum torque parameter (M_max_) and amount of energy dissipated in a single loading cycle (E_d_). Determining the equivalent stiffness value is a complex problem. The equivalent stiffness can be derived from potential energy and depends on damper movement direction. The detailed methodology of deriving the equivalent stiffness in systems with hysteresis can be found in [29]. In the initial stage of the loading and unloading process, M = f(Φ) characteristics follow a quasi-linear trend. The equivalent torsional stiffness parameter was defined as an average of k_eq_l_ and k_eq_u_ parameters. The process of determining the value of the parameters k_eq_a_, k_eq_l_ and k_eq_u_ for different underpressures is illustrated in Figure 8.

Experimentally-captured damping characteristics M = f(Φ) were recorded separately for the selected values of underpressure. Fragments of the characteristics from the range Φ < 1, 3 > (deg) and Φ < 5, 3.5 > (deg) were approximated with linear functions. The slopes of approximation lines were assumed as unknown k_eq_l_ and k_eq_u_ parameters defined as k_eq_i_ = dM/dΦ. The parameter k_eq_l_ value was estimated for a loading process (range Φ < 1, 3 > (deg)), similarly k_eq_u_ was defined using the unloading process (range Φ < 5, 3, 5 > (deg)). The region Φ < 0, 1 > (deg) was intentionally neglected to avoid possible disturbances related to grain reorganization in the changeable loading direction zone or technological clearances in the damper mounting point. The maximum torque parameter M_max_ was defined as a torque value measured for the Φ = 5 (deg) torsion angle.

The last parameter taken into consideration at this stage of research was related to the amount of energy dissipated in a single stabilized loading cycle. To define the E_d_ parameter value, the simplest numerical integration procedure based on the midpoint rule was used to calculate a numerical approximation for the area located inside experimentally captured dissipative characteristics.

The values of previously mentioned parameters were determined separately for each experimental probe conducted with a selected underpressure value. The obtained results are presented in Table 1.

Detailed analysis of collected experimental data revealed that the variation of the M_max_, k_eq_a_ and E_d_ parameters as a function of underpressure was nonlinear. Generally, the exponential upward trend for the VPP damper mechanical properties, during generation of higher partial vacuum values, has been discovered (Figure 9, Figure 10 and Figure 11).

The main objective of the experiments was to confirm the possibility of changing the energy dissipating properties of the VPP torsional damper prototype by controlling the underpressure parameter. Experimental data were used as the basis for the forthcoming modeling process, which was a classical inverse problem consisting of fitting the numerical model response into the real experimental results.

## 4. The Bouc–Wen Model

By analyzing the data depicted in Figure 6 it can be observed that the response of the VPP damper is nonlinear. In addition, the hysteretic type of behavior is confirmed. Recorded dissipative characteristics are generally symmetrical. Only in the case of atmospheric pressure (zero underpressure value, i.e., *p* = 0) was nonsymmetrical behavior observed (Figure 6a). Such a situation is predictable and has already been reported in the literature [30]. To assume VPPs as a continuous and homogeneous nonclassical solid body, a minimal underpressure threshold value must be applied. For example, in [31] it was shown that at least 0.005 MPa underpressure must be generated inside the granular core to provide a jamming mechanism. In [32] the authors examined a bended VPP beam and set the threshold partial vacuum value, enabling a solidification effect of the granular structure at 0.02 MPa. The modeling process of a fully loose granular material placed in a hermetic plastomer encapsulation is complex. It demands applying finite and discrete element methods together. Elastic membranes without tension caused by underpressure do not support grains in a determined way. Moreover, membranes do not exhibit explicit contact with the grains. In such a case the VPPs behave like a plastic mass that deforms locally rather than macroscopically. As a main part of the torsional damper the VPPs are expected to attenuate large amounts of energy. Consequently, the lowest ranges of underpressures are less important from an operational point of view.

As previously mentioned, the global behavior of VPPs and their controlling process in some aspects can be compared to magnetorheological fluids [33,34]. Expecting that in the near future VPPs could replace popular MR fluids in many engineering applications, especially in semi-active vibration damping, special attention was paid to rheological models of MRFs. Typically, the Bingham model is applied to model MRFs [35]. It is a comparatively simple model, making it easy to carry out numerical simulations. However, its biggest disadvantage is the impossibility for it to represent complex shapes of experimentally recorded hysteresis loops. Although there are many other, more complicated models, including the Gamota–Filisko [36], Jiles–Atherton [37,38] or Preisach hysteresis models [39], in the last decades the Bouc–Wen (B–W) model has become a common approach in hysteresis behavior modeling [40].

This model was initially proposed by Robert Bouc [27] and extended by Yi-Kwei Wen [41]. The complication level of the model is widely accepted in both computer simulation and control processes. One disadvantage of the B–W model is a relatively large number of parameters and the sensitivity of model responses to changes in their value.

The application of the B–W model is not only restricted to magnetorheological or electrorheological fluids. It is widely applied to capture the response of various nonlinear hysteretic systems. It has been modified and used to describe many engineering problems like base-isolation materials, multi-dimensional continuous systems, various viscous dampers, metals, masonry, timber and even transformers. It was also successfully applied in the structural control of nonlinear torsional dampers [42].

The mechanical scheme of the B–W model is depicted in Figure 12 and the restoring force is expressed as Equations (1) and (2) [28].

Variable *z* is given by the following formula:(1)$$F(t)=\frac{d^{2}x}{dt^{2}}+2\xi \omega_{n}\frac{dx}{dt}+\alpha {\omega_{n}}^{2}x+(1-\alpha )\omega_{n}z$$
(2)$$\frac{dz}{dt}=-\gamma z|\frac{dx}{dt}|{\left|z\right|}^{n-1}-\beta \frac{dx}{dt}{\left|x\right|}^{n}+A\frac{dx}{dt}$$
where
*F(t)* = force function;*ξ* = linear viscous damping ratio;*ω_n_* = pseudo-natural frequency of the system;*n* = degree of polynomial; and*α*; *γ*; *β*; *A* = parameters of hysteresis loop shape.

The model can be easily adapted to the rotational motion. Changing linear values to their rotational equivalents allows adoption of the B–W model to capture extraordinary features of the VPP torsional damper. Despite the compound form of the B–W model it can be represented in an analytical way. Equation (2) can be given by
(3)$$\frac{dz}{dt}=A-((\beta +\gamma \cdot \mathrm{sgn}(\frac{dx}{dt}))){\left|z\right|}^{n}$$

The parameter *n* is responsible for hysteresis loop radius of curvature [18]. Considering *n* = 2, the solution of Equation (3) has the following form:(4)$$z=\frac{\sqrt{A}}{\sqrt{\beta +\gamma }}\tanh(\sqrt{A(\beta +\gamma )}\cdot (x+C))$$

The constant *C* can be derived from boundary conditions.

Although the analytical solution with parameter *n* = 2 is not suitable to describe hysteresis loops obtained from experiments carried out on the VPP damper prototype, the authors decided to apply the numerical B–W model calibration process to fit the experimental data correctly.

## 5. Result Discussion

The B–W model in the version given by Equations (1) and (2) consists of seven coefficients. Generally, determination of the model parameters is based on laboratory test results and consists of various identification techniques. The most popular procedures are derived from heuristic optimization methods such as simulated annealing [43], particle swarm [44], firefly algorithms [45] or evolutionary algorithms [46]. Also, least-squares methods are frequently implemented, where the most popular are the Marquard–Levenberg or Gauss–Newton techniques [47].

In this preliminary approach to the modeling process the authors did not use a sophisticated numerical identification procedure. A simple Monte Carlo method was implemented to capture the model’s unknown parameter values. The applied technique consisted of a typical pattern: definition of the domain of possible inputs (the range of variability of parameter values of the Bouc–Wen model was taken from [31]), generation of random inputs over the uniform domain, carrying out simulation tests and comparing the model response with a real experimental data, and selecting the best solution. To provide a more accurate identification procedure, alternative sophisticated, nondeterministic optimization techniques are necessary. Genetic algorithms, simulated annealing or the swarm particles method can calibrate the discussed model with a better accuracy.

The numerical simulations of the VPP behavior were carried out for the B–W parameter values shown in Table 2. Table 2 also includes data reflecting the model accuracy, such as the coefficient of determination R^2^ and error percentage defined as a relative error.

The numerical behavior of the investigated VPP damper simulated by the Bouc–Wen model correctly reflects the real data. The lowest precision of the B–W model is observed for the 0.00 MPa underpressure (Figure 13a), which was expected and discussed in Section 4. The large and uncontrolled scattering of experimental data in this case is caused by unpredictable local deformations of loose grains not subjected to the partial vacuum. For the higher underpressure values, much better agreement and precision of the numerical response was noticed. The average value of the coefficient of determination is greater than 0.95. This leads to the conclusion that the proposed model can be applied to describe the VPP torsional damper behavior. Relatively high values of percentage errors are caused by the unsophisticated numerical method used for the model identification.

For the accurate modeling of the VPP controllable torsional damper using the B–W model it is necessary to consider the partial vacuum effects.

In the following part of the paper the proposition of the modified Bouc–Wen model for the granular damper is presented. The modification process consists of replacing the seven model parameters with the empirically defined underpressure functions. Taking into account the partial vacuum parameter, the initial B–W model for the investigated device can be redefined into B–W-p model defined as
(5)$$F(t)=\frac{d^{2}x}{dt^{2}}+2\xi (p)\omega_{n}(p)\frac{dx}{dt}+\alpha (p)\omega_{n}{\left(p\right)}^{2}x+(1-\alpha (p))\omega_{n}(p)z$$
(6)$$\frac{dz}{dt}=-\gamma (p)z|\frac{dx}{dt}|{\left|z\right|}^{n(p)-1}-\beta (p)\cdot \frac{dx}{dt}\cdot {\left|x\right|}^{n(p)}+A(p)\cdot \frac{dx}{dt}$$

Figure 14 depicts the verification process of numerical simulation results with real experimental data for representative values of underpressure.

Figure 14a–g presents graphical representations of the numerically determined parameters of the Bouc–Wen model for various partial vacuums.

General underpressure functions have been formulated on the basis of data presented in Figure 14. In the proposed approach, three various types of functions have been applied to capture the experimentally recorded behavior of the VPP torsional damper: an exponential rise to maximum three-parameter function y = y_0_ + α(1−e^(−β x)^), an exponential decay three-parameter function y = y_0_ + α e^(−β x)^ and an exponential growth three-parameter function y = y_0_ + α e^(βx)^. It has additionally been observed that the n parameter is underpressure independent (*n* = 1.48). Finally, the proposed functions depending on the partial vacuum describing the Bouc–Wen rheological model material constants are
(7)$$\omega_{n}(p)=\omega_{n0}+\alpha_{\omega n}\cdot e^{\left(-\beta_{\omega n}\cdot p\right)}$$
(8)$$\xi (p)=\xi_{0}+\alpha_{\xi }\cdot e^{\left(\beta_{\xi }\cdot p\right)}$$
(9)$$\alpha (p)=\alpha_{0}+\alpha_{\alpha }(1-e^{\left(-\beta_{\alpha }\cdot p\right)})$$
(10)$$\gamma (p)=\gamma_{0}+\alpha_{\gamma }\cdot e^{\left(-\beta_{\gamma }\cdot p\right)}$$
(11)$$A(p)=A_{0}+\alpha_{A}(1-e^{\left(-\beta_{A}\cdot p\right)})$$
(12)$$\beta (p)=\beta_{0}+\alpha_{\beta }(1-e^{\left(-\beta_{\beta }\cdot p\right)})$$
(13)n(p)=1.48=const

Introducing functions of the partial vacuum (p) into the initial B–W model causes evident complications. They are connected to the necessity of dealing with a number of additional unknown coefficients that must be defined when basing on laboratory test results. The minimum number of experiments that have to be carried out to find the empirical form of Equations (7)–(13) is three experimental series conducted for different underpressure values.

The modified B–W-p model in its final form includes 19 unknown parameters. This multiparameter B–W-p formula is the cost of the reliable modeling of the discussed controllable VPP torsional damper. It should be emphasized that such a cost seems to be reasonable and acceptable when VPPs are considered as an innovative and competitive alternative to a much more advanced group of smart structures.

## 6. Conclusions

In this paper an innovative prototype of controllable VPP torsional damper has been discussed. A novel idea of using jammed state granular materials has been presented. The biggest advantage of the presented device over the traditional group of torsional dampers is the possibility of controlling its dissipative characteristics in real-time. The VPP reaction time issue is one of the future fields of research. It depends on a wide group of parameters. Examples like automotive airbags indicate the possibility of quick responses of inflated and deflated devices. The next benefit resulting from the introduced approach is an economic aspect. The grains that are the main part of the damper can be taken from the wide group of recycled materials (tires or plastics). The manufacturing costs of VPP dampers can therefore be significantly reduced (by up to 100 times) in comparison to classical energy attenuators. The next advantage is a simple controlling mechanism. The mechanical properties of the VPP damper, like stiffness and amount of dissipated energy, can be rapidly changed by generating a partial vacuum inside the system. Even the simplest vacuum pump is an efficient device to generate the internal underpressure and to increase the damping properties of the investigated device. The granular jamming phenomenon is fully reversible and allows for rapidly turning the solidification mechanism on and off.

The experimental results obtained in the discussed work confirmed the previous results, published in the authors’ previous works, concerning simple cylindrical samples made of VPPs. Internal underpressure is a simple and convenient parameter that changes the global mechanical properties of the considered granular structures. As in the case of the uniaxial results, the discussed experimental data was modeled using the modified Bouc–Wen model. This paper confirms that generating higher underpressure value intensifies the grain locking mechanism resulting in increasing the dissipative properties of the VPP torsional damper prototype. The experimentally obtained dissipative characteristics of the VPP torsional damper are highly nonlinear. To capture extraordinary features of the device, a multiparameter rheological model has to be implemented. In this paper the authors chose the Bouc–Wen model of hysteresis. To extend the model enabling taking into account the controlling parameter (partial vacuum), the empirically determined underpressure functions have been introduced to the basic model equations. The efficiency of the modified B–W-p model has been experimentally verified and confirmed.

Further laboratory tests of granular materials subjected to the underpressure are intended. They will focus on investigating the influence of grain type and their microfeatures, like stiffness or roughness, on the macroscopic response of the VPP torsional damper. Prospective trends in the research of granular controllable dampers should also include the implementation of their numerical models into a finite element method code. Moreover, in order to improve the model prediction, more sophisticated model calibration methodologies have to be implemented.

## Figures and Tables

**Figure 1 materials-13-04356-f001:**
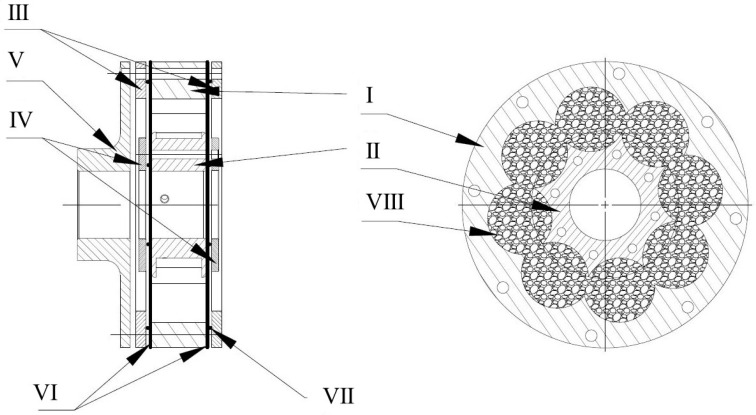
The scheme of the torsional controllable vacuum packed particle (VPP) damper prototype. (I) Outer driver, (II) inner driver, (III) outer rings, (IV) inner rings, (V) outer flange, (VI) sealing membranes, (VII) O-rings and (VIII) grains in the working chamber.

**Figure 2 materials-13-04356-f002:**
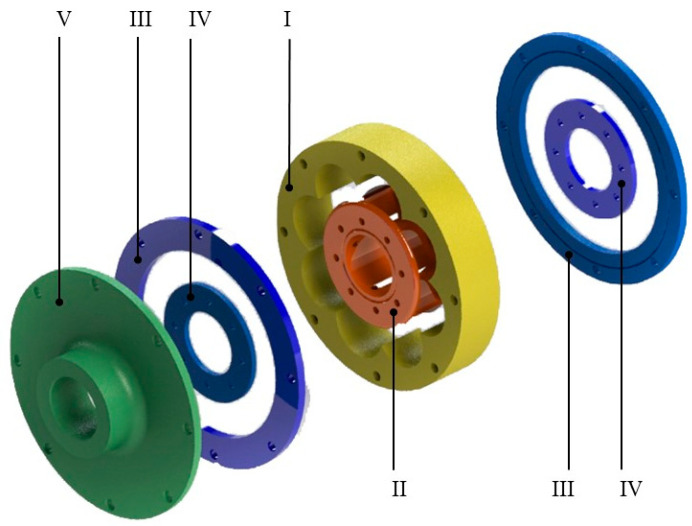
The CAD model of the investigated damper. (I) Outer driver, (II) inner driver, (III) outer rings, (IV) inner rings and (V) outer flange.

**Figure 3 materials-13-04356-f003:**
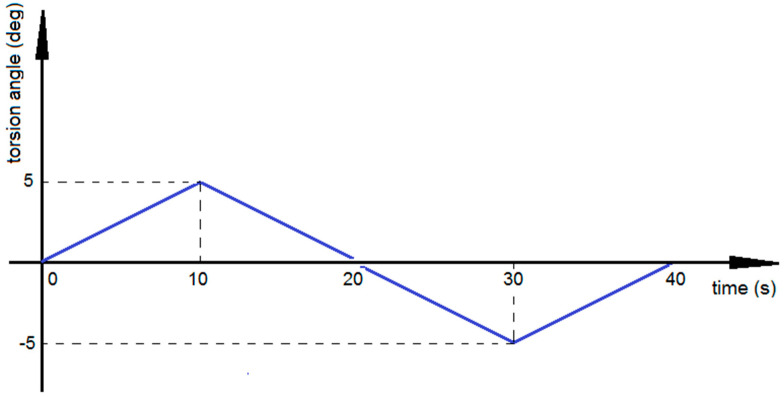
Single loading cycle.

**Figure 4 materials-13-04356-f004:**
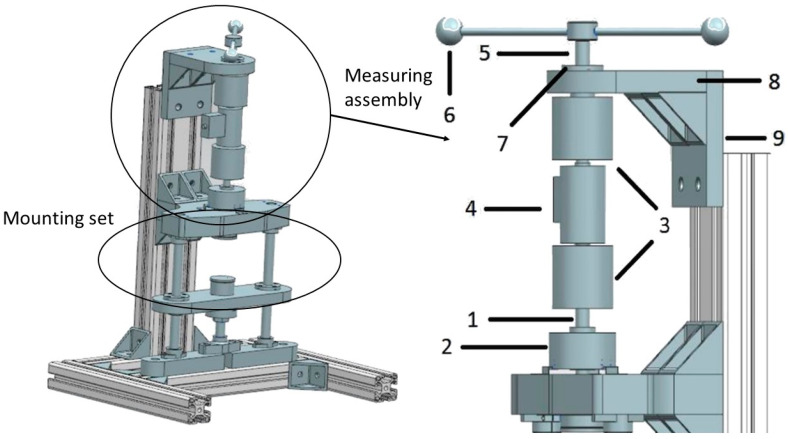
The test stand scheme. (1) Shaft, (2) encoder, (3) coupling sleeves, (4) torque transducer, (5) shaft, (6) lever, (7) thrust washer and (8,9) mounting plates.

**Figure 5 materials-13-04356-f005:**
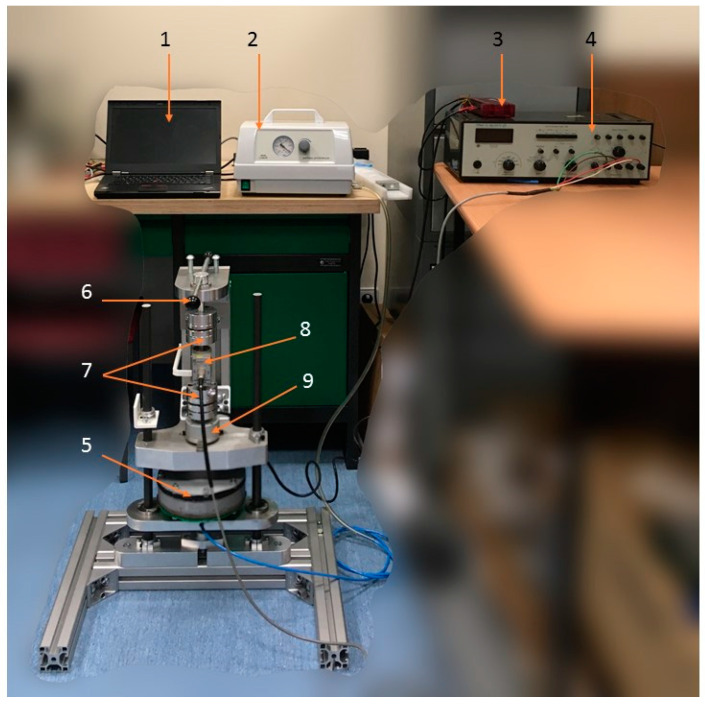
The test stand illustration. (1) PC, (2) vacuum pump, (3) data acquisition card, (4) signal amplifier, (5) VPP damper, (6) load lever, (7) coupling sleeves, (8) torque transducer and (9) encoder.

**Figure 6 materials-13-04356-f006:**
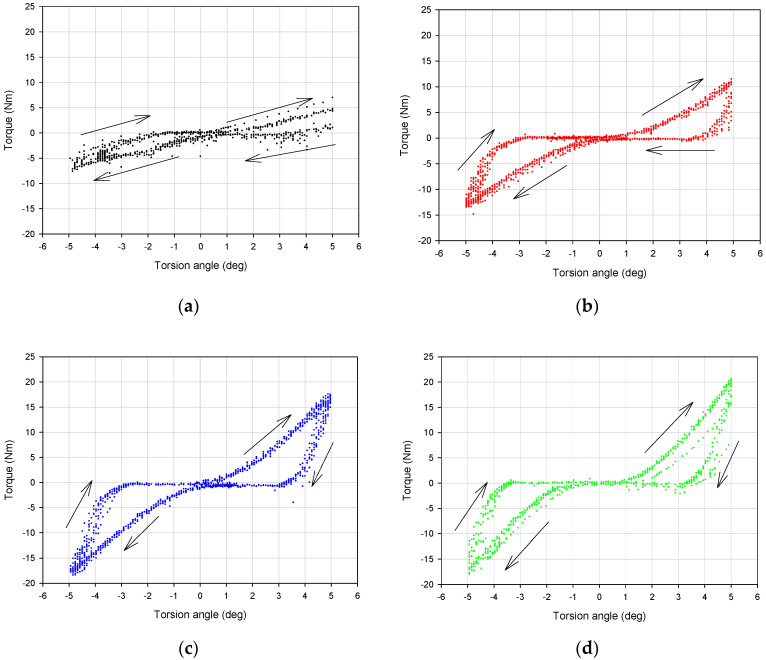
Damping characteristics for various underpressure values: (**a**) 0.00 MPa, (**b**) 0.01 MPa, (**c**) 0.02 MPa and (**d**) 0.05 MPa.

**Figure 7 materials-13-04356-f007:**
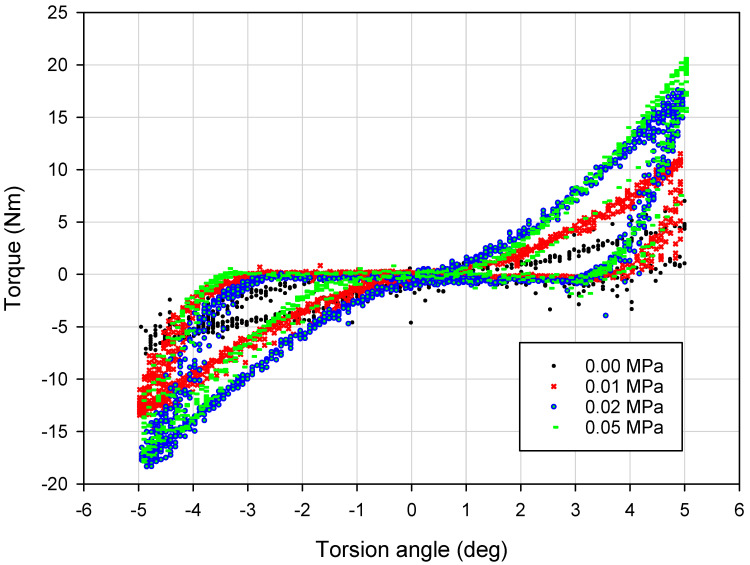
The influence of underpressure on damping properties of the VPP damper.

**Figure 8 materials-13-04356-f008:**
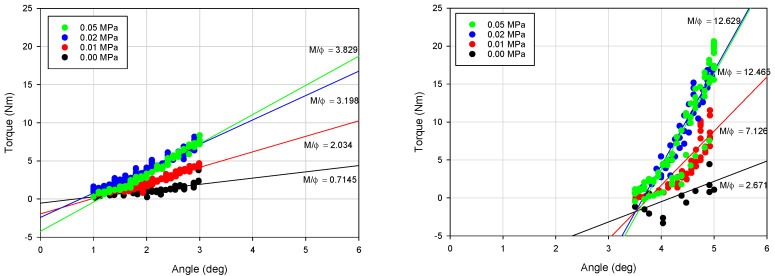
Determining the value of equivalent torsional stiffness parameters for various underpressures.

**Figure 9 materials-13-04356-f009:**
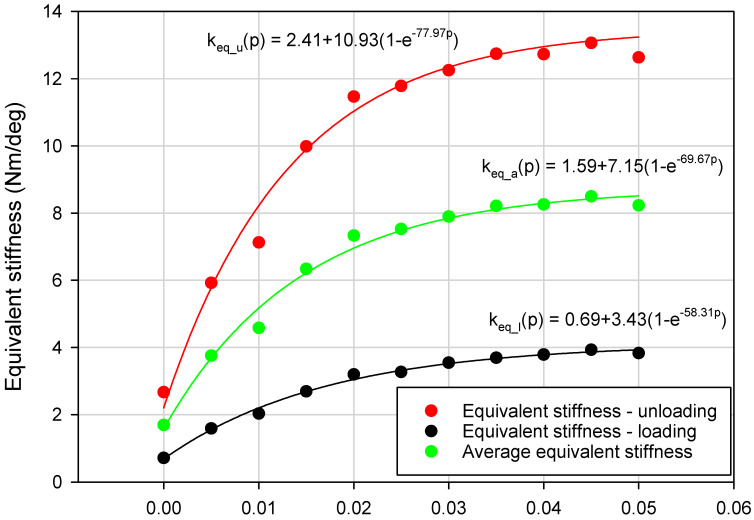
Exponential rise to maximum approximation for the k_eq_a_ parameter defined for various underpressures.

**Figure 10 materials-13-04356-f010:**
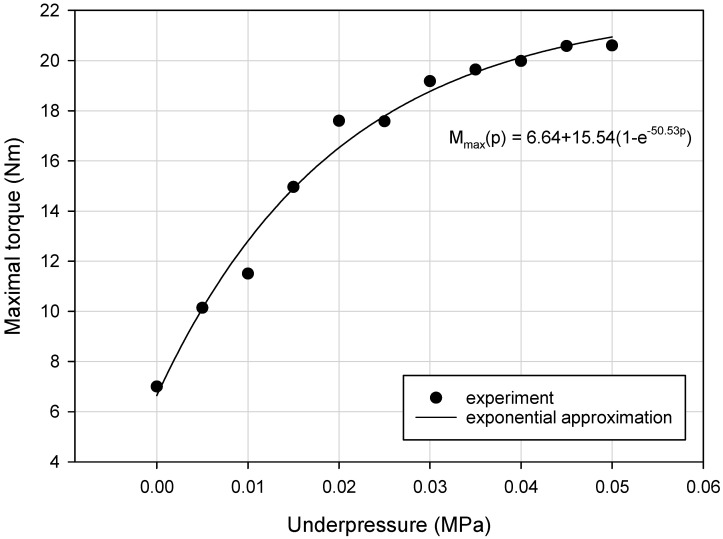
Exponential rise to maximum approximation for the M_max_ parameter defined for various underpressures.

**Figure 11 materials-13-04356-f011:**
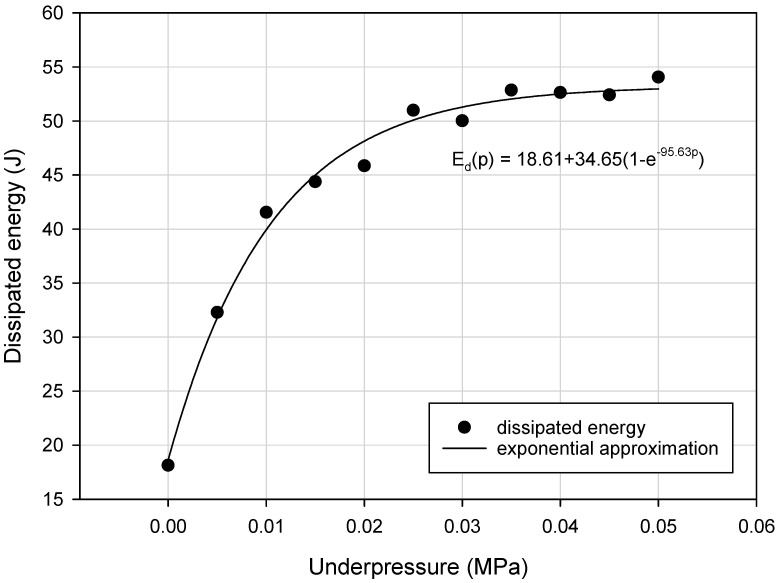
Exponential rise to maximum approximation of the E_d_ parameter defined for various underpressures.

**Figure 12 materials-13-04356-f012:**
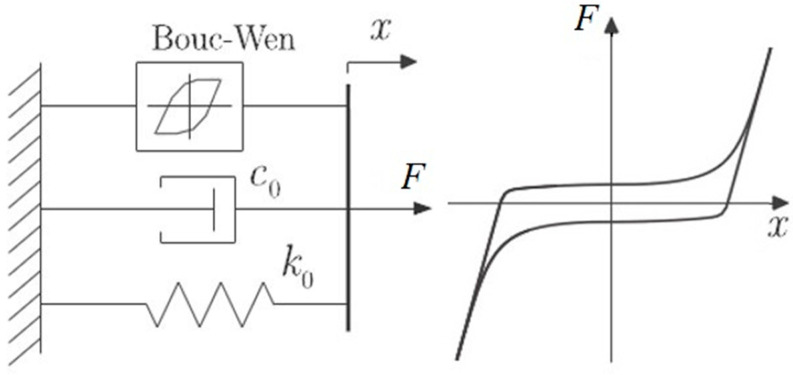
The scheme of the Bouc–Wen model.

**Figure 13 materials-13-04356-f013:**
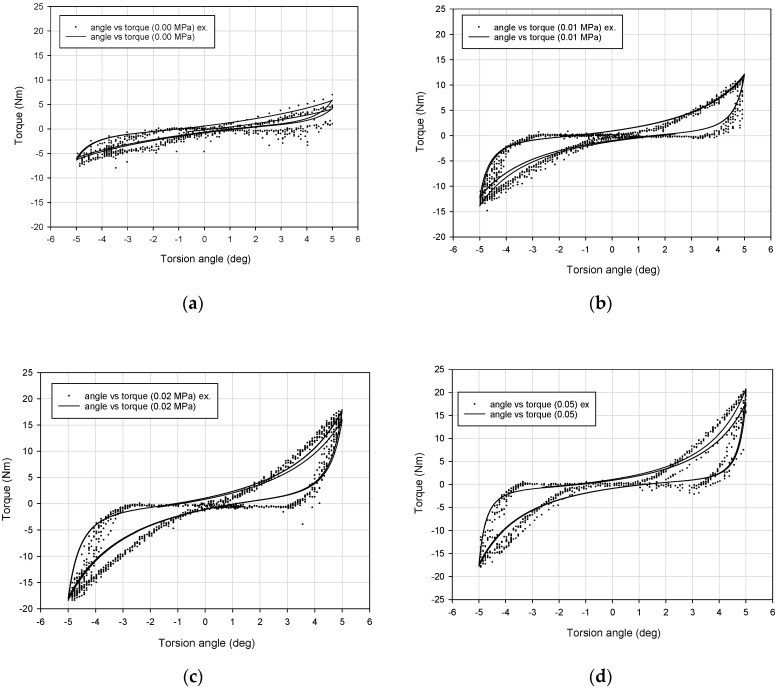
Verification of the numerical results and real VPP damper characteristics for representative probes for (**a**) 0.00 MPa, (**b**) 0.01 MPa, (**c**) 0.02 MPa and (**d**) 0.05 MPa values of underpressure.

**Figure 14 materials-13-04356-f014:**
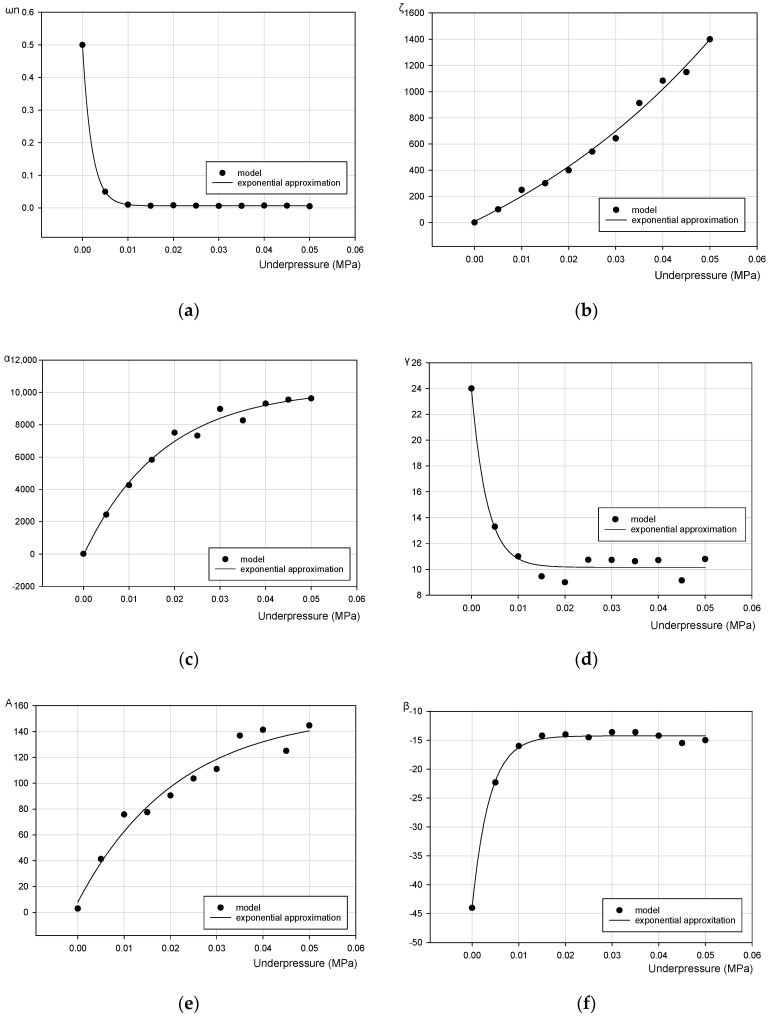
Variation of the Bouc–Wen model parameters in a function of partial vacuum. (**a**) ω_n_(p), (**b**) ξ(p), (**c**) α(p), (**d**) γ(p), (**e**) A(p), (**f**) β(p) and (**g**) n(p).

**Table 1 materials-13-04356-t001:** Selected mechanical features of VPP controllable torsional damper for representative underpressure values.

Underpressure (MPa)	Equivalent Loading Stiffness k_eq_l_ (Nm/deg)	Equivalent Unloading Stiffness k_eq_u_ (Nm/deg)	Equivalent Stiffness k_eq_a_ (Nm/deg)	Maximum Torque M_max_ (Nm)	Dissipated Energy (Nm/deg)
0.000	0.72	2.67	1.69	7.00	18.13
0.005	1.59	5.92	3.75	10.14	32.28
0.01	2.03	7.13	4.58	11.50	41.55
0.015	2.69	9.98	6.34	14.96	44.38
0.02	3.20	11.47	7.33	17.60	45.86
0.025	3.27	11.78	7.53	17.56	50.99
0.03	3.55	12.25	7.90	19.18	50.02
0.035	3.69	12.74	8.21	19.64	52.85
0.04	3.79	12.72	8.26	19.98	52.64
0.045	3.93	13.06	8.50	20.58	52.41
0.05	3.83	12.63	8.23	20.60	54.06

**Table 2 materials-13-04356-t002:** The Bouc–Wen model parameters.

p (MPa)	ω_n_ (1/s)	ξ (Nm∙s/°)	α (Nm/°)	γ (deg^−2^)	A (-)	β (deg^−2^)	n (-)	R^2^ (-)	Error (%)
0.000	0.5000	2	13	24.01	3.00	−44.00	1.48	0.857	19.2
0.005	0.0503	106	2877	12.88	36.19	−21.25	1.48	0.902	18.1
0.01	0.0100	250	4259	11.00	75.83	−16.00	1.48	0.931	14.7
0.015	0.0065	292	5646	10.98	77.54	−13.92	1.48	0.950	13.5
0.02	0.0080	400	7511	9.00	90.50	−14.00	1.48	0.962	13.3
0.025	0.0066	603	7401	9.05	101.40	−13.64	1.48	0.970	16.1
0.03	0.0065	720	9235	9.04	112.17	−14.78	1.48	0.976	15.6
0.035	0.0062	887	9797	10.43	118.76	−15.07	1.48	0.979	13.5
0.04	0.0064	992	8660	10.82	122.51	−15.07	1.48	0.981	13.4
0.045	0.0063	1280	9646	10.33	134.74	−13.62	1.48	0.983	12.9
0.05	0.0050	1400	9630	10.8	144.70	−15.00	1.48	0.984	12.2

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
