# Peer review of "Innovative Controllable Torsional Damper Based on Vacuum Packed Particles"

_materials, 2020, doi:10.3390/ma13194356_

Round 1
Reviewer 1 Report
Thanks to the authors for performing this research. The following comments need to be addressed:
- Like VPP defined in line 29, FDM needs to be defined in line 117.
- Same comment for ABS in line 135.
- Please provide a higher quality picture for Figure 3.
- Add gridlines for Figure 7 and make the axes range for all plots identical to be comparable.
- Gridlines are also needed for Figure 8 and all the other figures with plots.
- The figures with plots have already captions and axes titles. You should remove the plot titles.
- Provide reference for SigmaPlot software in line 196.
- Figure 15 plots are blur. Improve the quality if possible.
- In line 361 please revise this sentence “In this paper the authors chose a popular for such class of devices Bouc-Wen model”.
In summary, the manuscript is acceptable for publication upon addressing the above comments.
Reviewer 2 Report
Review of "materials-882175-peer-review-v1"
Summary: This paper describes the experimental testing and Bouc-Wen based modeling of a rotational particle damper with variable underpressure.
Feedback points:
- The abstract must be simpler: description of the device, description of the tests and modeling based on Bouc-Wen approach. All other sentences on “adaptive” damper etc to be removed because the paper only shows the characterization of the device, not the application to a benchmark structure.
- It is strongly recommended to put the description of the VPP in an additional section between “Introduction” and “Experiments” for better structure of the paper
- Section 2.2: why tests with triangular angle and therefore at constant tangential velocity? Reasons must be given.
- Figure 7 etc: include an arrow to show in the figure if the force first increases with increasing angular displacement or if it first goes along the flatter line.
- All torque-angle figures: same y- and x-scalings.
- Figure 8: make legend consistent with figure 7.
- The derivation of the equivalent stiffness must be described more precise: is k_eff the ratio between the torque and the angle of the torque branch when angle is increasing? Since the torque branches for increasing and decreasing angles are different the correct way to determine the equivalent stiffness is to determine numerically the elastic energy from angle=0 to angle=max and from angle=max to angle=zero; please see the given references for the computations of the equivalent stiffness (to be cited in the revised paper).
- Instead of dissipated energy the cycle energy equivalent viscous damping coefficient could be given.
- It must be mentioned that the Bouc-Wen model shows a torque-angle curve in the vicinity of the point (0,0) which does not pass through the point (0,0) and thereby the Bouc-Wen model cannot describe the measured torque-angle curve that does pass through point (0,0).
[1] Meirovitch L 2001 Fundamentals of Vibrations (New York: McGraw-Hill).
[2] Weber F. and Maślanka M. Precise Stiffness and Damping Emulation with MR Dampers and its Application to Semi-active Tuned Mass Dampers of Wolgograd Bridge. Smart Mater. Struct. 2014; 23: 015019..
Reviewer 3 Report
The paper presented a vacuum packed particle (VPP) based torsional damper and investigated its properties in quasi-static mode. Following comments should be addressed before the paper is accepted for publication:
- An important issue with the paper is the writing which should be revised to improve the understanding of the manuscript. There are several writing issues and poor word selection that should be corrected.
- Section 2.1.; page 4, lines 134-137: How have the type and size of the grains, as well as the volume fraction been selected?
- Figure 3: I am not sure if this figure is adding any information to the readers. It might be better to remove it.
- Page 5, Section 2.2: how have the test parameters, including the angle range and pressure range been selected. It is worthwhile to explain more about the rationale for selecting these ranges.
- Section 2.3: based on what is shown in Figures 5 and 6, the torque has been applied manually. If so, how have the tests been controlled to make sure that all of them are similar? For instance, how did the authors control the applied torque to follow the loading cycle presented in Figure 4?
- Section 2.3: It is worthwhile to further explain the data acquisition. Has any filter been used to avoid noises?
- Figure 7: hysteresis curves shown in Figure 7 are worth to be discussed more.
- How does the amplitude of the angle affect the responses of the damper? For instance, how are the hysteresis curves for Φ=±2◦ or Φ=±10◦?
- Page 8, line 188: What does “basic mechanical properties” mean? It is suggested to revise the sentence to avoid confusion.
- Page 8, lines 192-194: The initial stage in which a quasi linear trend is considered should be clearly mentioned.
- Page 8, lines 195-199: the paragraph is not clear and could not be followed. Please revise it.
- Figure 9: it is difficult to differentiate the trendlines that correspond to different pressures. Please use markers or different colors for each pressure to better distinguish the lines.
- Figure 10-12: It is recommended to add the equations of the exponential approximations shown in these figures.
- For all the figures the titles have been added above the figures which are basically the repeat of the caption. It is recommended to remove the titles.
- Page 12: References are required to be mentioned for Figure 13 and Eqs. (1) and (2).
- Page 12, lines 284-285: there is no “n” in Eq. (3) that has been addressed in this sentence.
- Table 2: Please add a column and show the error percentage and coefficient of determination (R2) between the model and the experimental results for each of the pressures.
- Figure 14: However, the model shows the same trend as the experimental data, the difference is considerably high even for higher pressures such as 0.02 Pa. How could this difference be explained?
- Page 16: coefficient of determination and average error percentage are required to be mentioned for the modified Bouc-Wen model (B-W-p).
- One of the main parameters affecting the dynamic performance of dampers is the driving frequency. How will the damper respond to higher driving frequencies? How should the model be modified to address the frequency?
- How does temperature variation affect the dynamic properties of the damper?
- Another main parameter to evaluate a controllable damper is its response time. What is the response time of the developed damper?
- Page 16, lines 347-349: It has been said that “The biggest advantage of the presented device … its dissipative characteristics.”. How is it the case comparing the proposed damper with for instance MR dampers, which have a response time of less than 0.001 sec? This comparison could not be justified without mentioning the response time of the VPP damper.
Reviewer 4 Report
In this paper, the authors design and investigate the controllable damper with vacuum-packed particles, and also the merits and demerits are discussed. There are a number of issues to be taken care of throughout the manuscript. Besides, the inclusion of some more information would strengthen the quality of the paper.
Here are my concerns:
- The novelty of this work is unclear. In particular, the authors should clarify the novelty of this work.
- [Abstract] should be more informative to present the findings of the work (including the results).
- [Introduction] Should be more organized and informative. The last paragraph of the introduction must be modified and updated with the novelty, scientific soundness, technical issues, and methods used, as well as the quantified results of the proposed work. The current content does not have any adequate information. It is better to outline the structure of the paper at the end of Section 1.
- “It consists of the outer driver (1), the inner driver (2), two outer rings (3), two inner rings (4), an outer flange (5), two sealing membranes (6), two O-rings (7) and a working chamber with granular material (8)”. It's better to use “Roman Numerals” (I… VIII) instead of (1), (2), (3).
- The authors should clearly label of prototype schematic in the figures (Fig. 1, Fig. 2…) with each component.
- Figures 10, 11, and 12 no need to separate. Please make one figure.
- The figure quality needs to be improved in this manuscript (Like Fig. 14….).
- The authors should check the X-axes value in Figure 10 and Figure 15.
- The authors should include the comparison results with previously reported works.
- [Conclusion] should summarize the key idea and theory.
Reviewer 5 Report
In this paper, the authors presented a novel approach in application of Vacuum Packed Particles (VPP), which is the controllable torsional damper. Because it is supported on laboratory tests the subject of the paper is interesting for publishing.
There are 44 references in the text, being about 41% from the last 5 years, and about 36% are more than 10 years old, which is a bit high but acceptable.
There are some English grammar issues to correct, so a careful revision of the manuscript should be carried out. For example, in line 29 the sentence “that pretend to a group of materials” does not make much sense. Or in line 196, where is “were approximated in a SigmaPlot software”, probably the authors meant to say that the SigmaPlot software was used to carry out the task. It is also not very clear what authors meant to say with the sentence “model reveals the greatest interest of scientists and researchers” (line 256).
In lines 53, 96, 250 and 251, the authors referred the acronym MR. However, there is no explanation about it in the text. Earlier it is presented MRF (magnetorheological fluids), which is probably what the authors meant to write.
A similar problem is presented in line 117, because no explanation is presented for FDM. Are the authors referring to Fused Deposition Modeling (FDM)? Is so, was the damper prototype created with the aim of a 3D printer?
In lines 75-76 the authors stated that Genetic Algorithms were used to calibrate the Bouc-Wen model, but it is not clear if the procedure was really adopted in the present study (later on in the paper it seems not…). If so, how was this done?
Despite being possible to compute the loading velocity of the laboratory tests, using the values presented in Figure 4, I believe that the authors should present that information in the text.
The legends of Figures 5 and 6 should be completed, adding an explanation for the presented numbers (despite that information is presented in the text), because it will help any reader that founds those figures in the Internet during a search.
There is a lack of references to support what it is stated in lines 264 to 266.
It is not very clear how the Monte Carlo method was used to capture the Bouc-Wen model parameters. Were the random variables considered having uniform distribution?
In Table 2, only the underpressure units are presented which should be completed, namely adding the units of the pseudo-natural frequency, and clarifying if the linear viscous damping ratio is presented in percentage, for example.
The authors presented in the conclusions the adjective “interesting”. I believe that it should be the reader to reach that conclusion, not the authors…
For all those reasons, I believe that the paper is only acceptable for publishing after a revision.
Round 2
Reviewer 2 Report
the paper is revised accord. to the review ==> accepted
Author Response
Dear Reviewer,
Thank you for your previous comments and acceptance of the revised paper.
Yours faithfully,
Dominik Rodak, Robert Zalewski
Reviewer 3 Report
- The manuscript has been revised using the Track Change, which made it crowded and incomprehensible. The Track Changes should be applied and the final version should be sent for review. In the current form, it is not possible to review the manuscript and track the comments. Moreover, the corresponding changes to each comment should be highlighted in the manuscript and addressed in the letter (page and line numbers), so they could be found in the manuscript.
Regarding the answers that have been provided in the cover letter, the following comments should be considered:
- There are plenty of writing issues in the letter that makes it difficult to understand. The response letter is needed to be revised. For instance, Answers to the third referee, Comment #2: It is not clear what the authors mean on “The presented in the paper experimental results are the next stage of investigations on Vacuum Packed Particles.”.
- Answers to Comments 5 and 19: It is not clear what do the authors mean about “Mentioned in ad. 4” and “The answer is mentioned in ad. 17.”. If they address to the answers to Comments 4 and 17, these answers look not be related to Comments 5 and 19.
- Discussion on Comments 2, 4, 5, 6, 8, 18, and 23 are required to be added in the manuscript.
Reviewer 4 Report
The paper quality is improved after the revision.Author Response
Dear Reviewer,
Thank you for your previous comments. We hope that the revised paper is acceptable now.
Yours faithfully,
Dominik Rodak, Robert Zalewski
Reviewer 5 Report
The manuscript has been much improved.
In general, the authors have addressed most of my concerns about the paper.
In my pdf version, the title of the horizontal axis of the first chart of Figure 9 is cut, which should be corrected. Most probably, it is “Underpressure [MPa]”.
